# Effect of Three Bakery Products Formulated with High-Amylose Wheat Flour on Post-Prandial Glycaemia in Healthy Volunteers

**DOI:** 10.3390/foods12020319

**Published:** 2023-01-09

**Authors:** Claudia Di Rosa, Elisa De Arcangelis, Virginia Vitelli, Salvatore Crucillà, Martina Angelicola, Maria Carmela Trivisonno, Francesco Sestili, Emanuele Blasi, Clara Cicatiello, Domenico Lafiandra, Stefania Masci, Maria Cristina Messia, Laura De Gara, Emanuele Marconi, Yeganeh Manon Khazrai

**Affiliations:** 1Research Unit of Food Science and Human Nutrition, Department of Science and Technology for Humans and the Environment, Università Campus Bio-Medico di Roma, Via Alvaro del Portillo, 21-00128 Rome, Italy; 2Dipartimento Agricoltura, Ambiente e Alimenti, Università degli Studi del Molise, Via F. De Sanctis snc, 86100 Campobasso, Italy; 3University Hospital of Verona, Piazzale Aristide Stefani, 1, 37126 Verona, Italy; 4Department of Agriculture and Forest Sciences, University of Tuscia, Via San Camillo de Lellis snc, 01100 Viterbo, Italy; 5Department for Innovation in Biological, Agro-Food and Forest Systems, University of Tuscia, Via San Camillo de Lellis snc, 01100 Viterbo, Italy; 6Centro Interateneo di Eccellenza per la Ricerca e l’Innovazione su Pasta e Cereali trasformati (CERERE), 86100 Campobasso, Italy; 7Operative Research Unit of Nutrition and Prevention, Fondazione Policlinico Universitario Campus Bio-Medico, Via Alvaro del Portillo, 200-00128 Rome, Italy

**Keywords:** biscuit, bread, high-amylose flour, glycaemic index, resistant starch

## Abstract

Both Glycaemic index (GI) and Glycaemic Load (GL) were introduced to measure the impact of a carbohydrate-containing food on blood glucose. From this perspective, high-amylose (HA) flours, with a higher percentage of resistant starch (RS), may represent a suitable raw material to improve the glycaemic response. The present work aims to investigate the GI of HA bakery products (biscuits, taralli and bread) compared to products obtained from conventional flour. Ten healthy volunteers were enrolled and their capillary blood glucose was measured every 15 min for 2 h after the consumption of HA and control products containing 50 g of available carbohydrates. On average, in the three bakery products, the amount of total starch replaced by RS was equal to 12%. HA biscuits and HA bread showed significantly lower GI than their control counterparts (*p* = 0.0116 and *p* = 0.011, respectively) and better glycaemic control. From the survey to assess liking and willingness to pay on HA snacks, HA packages received an average premium of €0.66 compared to control products. Although HA flour results in lower GI in both biscuits and bread, further studies are needed to evaluate the correct composition of HA products to have beneficial effects on post-prandial glycaemia.

## 1. Introduction

According to the International Diabetes Federation (IDF), 536.6 million people suffer from diabetes (both type 1 and type 2) and 6.7 million people died from these diseases or their complications in 2021 [1]. Type 2 diabetes (T2D) represents 90% of the diagnoses of diabetes [1,2]. Its causes are not fully understood but a strict relationship with obesity, obesogenic environment, increasing age and ethnicity exists [1]. Thus, T2D, as well as elevated post-prandial glycaemia, could be prevented by correct management of glucose metabolism and obesity through lifestyle changes comprising both physical activity and healthy food choices [1]. Among all macronutrients, carbohydrates are those which most affect glycaemia [3]. In the last decades, the role of the amount and quality of carbohydrates-containing foods as a risk factor for the development of T2D has been investigated [4]. In 1981, the concept of Glycaemic index (GI) of foods was introduced by Jenkins et al. to classify not only foods rich in simple sugars but also starchy foods [5]. GI is considered as the incremental area under the curve (IAUC) expressed as a percentage after the consumption of 50 g of available carbohydrates and it is used to rank carbohydrate-containing foods according to the rate of sugar digestion, absorption and metabolism [6]. The GI is classified as high, medium and low when it is higher than 70, between 55 and 69 or lower than 55, respectively [7]. High-GI foods cause a rapid rise in glucose in the bloodstream, reducing insulin efficiency over time, due to the downregulation of insulin receptors [8]. Conversely, a lower GI diet, rich in dietary fibre, may prevent diabetes and insulin resistance [9] counteracting the onset of cardiovascular diseases [8,10]. GI does not consider the portion size of foods, and for that reason the Glycaemic Load (GL) has been introduced. It is calculated as the product between the GI and the amount (in grams) of available carbohydrates in a food portion size/100 [11]. The GL is classified in high (above 20), medium (11–19) and low (under 10). This is an additional index that is related to the food portion size; thus, a smaller portion size of a high GI food could also be consumed without a dramatic impact on glycaemia [12]. Nevertheless, still little association was found between total carbohydrate intake or simple sugars intake and the onset of T2D. Livesey et al. published a meta-analysis in which they assessed that high-GI and GL diets were strongly associated with the onset of T2D, even in healthy subjects [13]. Unfortunately, these two parameters are not always considered by health professionals because they still believe that complex carbohydrates do not have the same effects on glycaemia as simple sugars. Some carbohydrate-containing foods such as bread, potatoes or rice [14] have a high GI, and thus the impact on glycaemia could be similar to the one obtained after the ingestion of simple sugars-containing foods [3].

Wheat flour is mainly composed of readily digestible carbohydrates, in the form of starch which is structured as amylose and amylopectin. Amylose is a linear polymeric chain of (1→4)-linked α-D-glucopyranosyl units, while amylopectin has a branched structure of α-D-glucopyranosyl unit with both (1→4) and (1→6)-α linkage. In conventional wheat, these macropolymers are organized as 25% amylose and 75% amylopectin. On the other hand, high-amylose genotypes are characterized by a higher amylose proportion (>70%) [15,16,17], obtained through the manipulation of gene expression coding for enzymes normally implicated in starch synthesis (Granule bound starch synthase, starch synthase, starch branching enzyme) [15]. In particular, silencing of the genes encoding starch branching enzyme SBEIIa results in a phenotype with a strong increase in amylose content (>70% of total starch) and drastically modifies starch granule morphology. These wheats provide a marked increase (up to 10%) of the resistant starch fraction [18]. Resistant starch can be defined as a portion of starch that remains undigested in the small intestinal tract, and reaches the large bowel, where is fermented by the gut microbiota, with multiple beneficial effects on human health [19,20] associated with metabolic products released during its fermentation, particularly the production of the short-chain fatty acids (SCFAs) butyrate, acetate and propionate. The mechanisms behind the reduced digestibility of amylose-rich starch include multiple intermolecular interactions, formation of amylose–lipid complexes, and modified starch pasting properties [19]. In addition, a health claim on reduced postprandial glycaemia on resistant starch is enclosed in the EU Reg. 432/2012 [21], when this fraction is at least 14% of total starch. Different authors evidenced the health-promoting potential of high-amylose wheats due to the ability to reduce the GI of pasta and baked products [22,23,24,25,26].

Biscuits are consumed by all age groups because of the long shelf-life, low cost and sensorial properties. They are normally manufactured with flour, sugars and fats and can be consumed as part of an everyday diet, for example during a sweet breakfast. Numerous attempts have been made to reduce the GI of biscuits; for instance, [27] Di Cairano et al. produced biscuits using non-conventional flours, made of pseudocereals and legumes. Similarly, in another work, the addition of resistant starch and the replacement of sucrose caused a substantial decrease in the in vitro glycaemic index of experimental biscuits [28]. “Taralli” is a typical Italian savoury bakery product that can be consumed as a snack. It is normally produced in South Italy, employing bread wheat flour, salt, water, olive oil, herbs and/or spices, but also white wine in some cases. Attempts to enhance the nutritional properties of taralli referred to an improved composition (i.e., the polyphenolic compounds content) have been reported [29], while no data are available on the possibility to reduce the GI. Bread, on the other hand, is a more popular wheat-based product that other works have already formulated with high-amylose wheats [30,31]. Indeed, Hallström et al. [30], found that bread with elevated amylose content improved the glycaemic response but not the insulinemic response after a meal. In vitro starch hydrolysis of bread produced with high-amylose flour was measured by Li et al. [31], reporting a substantial reduction of digestion rate compared to the control.

The aim of the present work is to investigate the properties of high-amylose wheat baked products (biscuits, taralli and bread) by means of in vivo evaluation of glycaemic index in healthy individuals, by comparing to control products obtained with regular flours and to test appreciation and willingness to pay for high-amylose biscuits and taralli, compared to the controls.

## 2. Materials and Methods

### 2.1. Flour Samples

High-amylose wheat grain (*Triticum aestivum* L.) cv Cadenza [32] was grown at the Experimental Farm of University of Tuscia and milled to flour (HA Flour) by Grandi Molini Italiani SpA (Venezia, Italy). A control flour was obtained by mixing commercial type 1 flour (ash = 0.70% d.w.) and type 00 flour (ash = 0.50% d.w.) in a 65%/35% proportion to obtain a similar chemical composition of the experimental flour, with the exception of amylose and resistant starch content (Table 1). All the other ingredients were purchased in local markets.

### 2.2. Products Development

High-amylose foods and the respective control foods contained the same amount of net energy and carbohydrates (50 g) in terms of fibre, but not resistant starch, in order to evaluate the metabolic response consistently with the health claim, enlisted in Reg. EU 432/2012 [21], that prescribes the substitution of digestible starch with resistant starch.

#### 2.2.1. Biscuits

Shortbread biscuits were produced by mixing eggs (28 g/100 g flour) and sucrose (30 g/100 g flour) in a planetary mixer (Kenwood Chef XL mod. KVL60, Kenwood Ltd., Havant, UK), then flour and chemical leavening agents (3 g/100 g flour) were added, together with sunflower oil (30 g/100 g flour) and water (5 mL/100 g flour). The dough was mixed (15 min) allowed to rest (10 min) and then extruded (Kenwood Ltd., Havant, UK). The biscuits were baked on an aluminium tray in a rotatory electric oven (CIMAV, Villafranca, Italy) at 180 °C for 20 min. After cooling, biscuits (approx. length: 5 cm, width: 2.5 cm, height: 1 cm) were milled with a refrigerated miller for analysis.

#### 2.2.2. Taralli

Taralli were produced according to a homemade recipe, without the use of white wine to avoid dislike among participants. Flour, extra virgin olive oil (20 g/100 g flour), salt (1.5 g/100 g flour) and water (64 mL/100 g flour for high-amylose taralli and 45 mL/100 g flour for control taralli) were added in a planetary mixer (Conti, Bussolengo, Italy mod, SP 20 2V). The dough was mixed for 15 min and then allowed to rest for 20 min. The dough was manually shaped rounded to form taralli (approximately diameter = 3 cm) and cooked in boiling tap water until it rose to the surface. After draining the excess of water, the taralli were baked on an aluminium tray in a rotatory electric oven (CIMAV, Villafranca, Italy) at 180 °C for 35 min. Samples were milled with a refrigerated miller for analysis.

#### 2.2.3. Bread

Bread (straight dough method) was produced by mixing flour, salt (1.5 g/100 g flour), dry baker’s yeast (0.7 g/100 g flour) and water (80 mL/100 g flour for high-amylose bread and 60 mL/100 g flour for control bread) in a planetary mixer (Conti, Bussolengo, Italy mod, SP 20 2V). The first proofing was conducted for 1 h at 30 °C. The dough was put in an aluminium container and left for the second proofing for 150 min at 30 °C. The dough (370 g) was baked in a static electric oven (CIMAV, Villafranca, Italy) at 230–180 °C for 1 h. The sample was freeze dried for analysis, except for resistant starch content that was determined on the fresh product.

### 2.3. Chemical Analysis

Moisture and ash content were determined according to ICC 109/1 and 104/1 respectively (ICC, 1995 [33]). Total fat was analysed by acidic hydrolysis (ICC method 136), while protein content was evaluated according to the Dumas combustion method, AACC method 46-30 (AACC, 2000 [34]) using a Leco nitrogen determiner, model FP 528 (Leco Corp., St. Joseph, MI, USA). The analysis of total starch (AACC method 76-13.01), amylose and resistant starch (AACC Method 32-40.01) were carried out using enzymatic assay kit (Megazyme International Ltd., Wicklow, Ireland). Sugar content in biscuits was calculated as sugar added according to the recipe.

### 2.4. Study Population

Ten healthy volunteers were recruited for this study at Campus Bio–Medico University of Rome. Participants were enrolled according to the following inclusion and exclusion criteria. Inclusion criteria were: (1) age between 18 and 30 years to reduce inter-individual glycaemic variation; (2) Body Mass Index (BMI) comprised between 18.5 and 24.9, 3) who had signed informed consent. Exclusion criteria included subjects who: (1) were involved in other studies, (2) used medications, (3) had allergies (included asthma), (4) had organic (inflammatory, neoplastic, metabolic diseases) or psychiatric diseases (anorexia, bulimia, depression), (5) were coeliac or gluten-sensitive, (6) had type 1 or 2 diabetes mellitus or used medications that could impair glucose metabolism, (7) had kidney failure (creatinine > 1.5 mg/dL) or liver failure (ALT and AST twice more than normal values), anaemia (Hb < 12 g/dL) and other chronic conditions. At the end of the enrolment phase, 5 men and 5 women were enrolled.

The mean anthropometric parameters of participants at baseline are reported in Table 2.

All investigations were conducted following the Declaration of Helsinki. The study was approved by University Campus Bio–Medico di Roma Ethical Committee and all the subjects signed an informed consent statement allowing their anonymized information to be used for data analysis. Participants’ records were anonymized and de-identified before the analysis.

#### 2.4.1. Anthropometric Parameters

To measure anthropometric parameters (weight and height), a SECA 700 scale with integrated altimeter was used. Participants were measured only at T0 for the enrolment phase. BMI was calculated through the ratio of body weight (kg) over height squared (m^2^).

#### 2.4.2. Glycaemia Measurement

A glucometer with pen and reactive stripes was used to measure the capillary glycaemia after food or glucose ingestion every 15 min for 2 h. As standard, glucose was used for the first and last measurements.

### 2.5. Study Protocol

During this study, participants followed their habitual diets and they were called to measure their glycaemic values with a glycaemic curve after the ingestion of foods. Participants were blinded; thus, they did not know if the food they ingested was a HA product or a control one.

At T_0_ participants were submitted to: (1) anthropometrical measurements, (2) medical visit with anamnesis paying attention to diseases or intake of medications, (3) nutritional anamnesis.

On the first and the last day of the study, participants consumed 50 g of glucose in 250 mL of water, while on other days they ingested control and HA foods on an empty stomach in a randomized order every 3 days (with two days of wash out). On the test day, fingerstick blood glucose was measured while fasting and every 15 min for 2 h after food consumption.

Subjects were not allowed to drink alcohol or to practice physical activity in the 72 h before the test; they were also invited to standardize the dinner before the test by consuming a light meal, avoiding the consumption of fibre-rich–foods (such as legumes and whole cereals).

### 2.6. Consumer Test Questionnaire

During the product tasting (biscuits and taralli), all participants filled out a pilot questionnaire designed to check potential differences in organoleptic characteristics, general liking and willingness to pay for HA and control products (see Appendix A).

The survey consisted of both a blind and informed condition on the product type (control vs. HA) and nutritional properties of HA snacks. In the blind condition, survey participants rated their overall liking and selected the appropriate sensory attributes (colour—yellow, brown; size—small, large, thin, thick; taste and flavour—crumbly, chewy, crispy, doughy, dry, stringy, fatty, salty, sweet, fragrant, toasted, bitter) on a 9-point scale ranging from “dislike a lot” to “like a lot”.

At the end of the blind sensory test, two questions were asked about willingness to pay (WTP). The first question referred only to the product tasted: “*How much would you be willing to pay for a snack pack of the product you just tasted*?”. WTP’s second prompt was given after the information was disclosed. The characteristics concerning the health properties of high-amylose flour, which was used for the production of biscuits and taralli, were summarized with the message ‘One of the products you have tasted contains natural high-amylose flours, which raise blood sugar less than a normal product’, followed by the WTP question ‘*How much would you be willing to pay for a snack pack of products containing natural high-amylose flours*?’. In order to obtain further indications on the purchase intention of the new product, two final questions were asked: “*If you found this type of product on the market, how likely would you be to buy it*?” and “*Would you recommend the purchase of the product with high-amylose flours*?”. These questions were rated along a noncomparative rating scale with 4 options.

## 3. Statistical Analysis

The statistical analysis was conducted through the statistical program GraphPad Prism version 9.4.1. The area under the curve (AUC) was calculated for both high-amylose and control foods through integral calculation and for glucose (taking in consideration the mean values between the first and the last measurement) and then the GI was calculated as the percentage ratio between the two areas: glycaemic index = (food area/glucose area) × 100 [35,36]. Moreover, a one–way ANOVA with Tukey’s post hoc test was conducted to evaluate the differences at each time between high-amylose food, control food and glucose. A *p* value < 0.05 was considered to be statistically significant.

The scores obtained for the level of liking were analysed via the XLSTAT software with an ANOVA analysis, with the aim of investigating the role of the different attributes in the overall liking perception, and to test whether the participants’ liking of the HA and control products significantly differed. The differences between the hypothetical value (euro per single-serving snack) expressed by the panel at the end of the test, before and after the disclosure of information on the effect of HA flour on health, were tested with the paired-sample t-test.

## 4. Results

### 4.1. Properties of Baked Products

The average chemical composition of high-amylose and control products is reported in Table 3.

Sugars that were calculated according to the recipe totalled approximately 19% of the biscuits. Remarkably, the resistant starch content is more than 10 times higher in experimental biscuits, due to the higher amylose content of the starting raw material. Overall, the resistant starch fraction accounts for 7.9% of the carbohydrates normally considered as digestible (total starch and sugars), while in control biscuits it is <1%. Resistant starch (% total starch) is decreased by 31% after processing. The product cannot claim the health effect on postprandial glycaemia, but a consistent proportion (11.7%) of digestible starch is replaced by a non-digestible polysaccharide compared to a conventional product.

The chemical composition of high-amylose and control taralli was, overall, comparable, except for RS that reached 7% in high-amylose taralli and 0.7% in control products. In total, resistant starch reached 12.5% of total starch, not being able to reach the cut off values for the health claim on resistant starch.

High-amylose bread reported a very similar starch composition compared to taralli. Furthermore, in this case, RS reached 12.5% of the total starch, not bearing the health claim on resistant starch. Nevertheless, in this case the control product accumulated a higher RS fraction compared to the starting raw material.

### 4.2. Biscuits

From the glycaemic curve, we observed significant differences between HA biscuits and glucose at 15′ (*p* = 0.0036), 30′ (*p* < 0.0001), 45′ (*p* < 0.0001), 60′ (*p* = 0.0001), 75′ (*p* = 0.0102) while between HA biscuits and control biscuits, there were no significant differences at each time. We also found statistical differences between control biscuits and glucose at 30′ (*p* = 0.0001), 45′ (*p* < 0.0001), 60′ (*p* = 0.0035). At 45′ of the glycaemic curve, we observed the maximum peak for HA biscuits, for control biscuits and for glucose at 110.1 mg/dL, 118.8 mg/dL and at 158.6 mg/dL respectively (Figure 1). The glycaemic index was calculated as 84.28 ± 5.21 and 90.29 ± 6.15 respectively for HA biscuits and for control ones (*p* = 0.0116), and in fact, HA biscuits showed a lower area under the curve (810.8 ± 37.37) than control biscuits (870.15 ± 71.34) (*p* = 0.0125).

We also calculated the GL of HA and control biscuits. Considering the portion of 30 g [37] and that in 100 g there were 58.3 g of available carbohydrates, we estimate that in the portion there were 17.5 g of carbohydrates, thus GL was calculated as: (84.28 × 17.5)/100 = 14.75 (medium glycaemic load). Instead, control biscuits had a GL of 16.77 (medium glycaemic load) considering 61.9 g of available carbohydrates in 100 g of product, thus 18.57 g in 30 g (portion size).

### 4.3. Taralli

From the glycaemic curve, we observed significant differences between HA taralli and glucose at 15′ (*p* = 0.0217), 30′ (*p* < 0.0001), 45′ (*p* < 0.0001), 60′ (*p* = 0.0006), 120′ (*p* = 0.0228) while between HA taralli and control ones, there were no significant difference at each time. We also observed significant differences between control taralli and glucose at 15′ (*p* = 0.0004), 30′ (*p* < 0.0001), 45′ (*p* < 0.0001). For taralli, from the glycaemic curve, we observed that the maximum peak for the HA product was of 116.6 mg/dL after 45 min, while for the control one, it was of 127.1 mg/dL after 60 min. The maximum glycaemic peak of glucose was always of 158.6 mg/dL after 45 min (Figure 2). The glycaemic index was calculated as 90.32 ± 5.9 and 91.73 ± 8.32, respectively, for HA taralli and for control ones. These two GIs are not statistically different, in fact the area under the curve of both products was similar (868.25 ± 27.69 and 880.9 ± 44.46 respectively for HA taralli and control ones) without significant differences.

The GL was calculated in the same way as for biscuits. Thus, considering that in 100 g of HA taralli there were 55 g of available carbohydrates, we calculated that the GL of a portion of taralli (30 g) [37] was 14.8 (medium glycaemic load). For control taralli, the amount of available carbohydrates in 100 g was 55.8; thus, the GL was 15.2 (medium glycaemic load).

### 4.4. Bread

Comparing each time point of the glycaemic curve, we observed statistical differences between high-amylose bread and glucose at 15′ (*p* = 0.0136), 30′ (*p* = 0.0021), 45′ (*p* < 0.0001), 60′ (*p* = 0.0006) and between control bread and glucose at 15′ (*p* = 0.0158), 45′ (*p* = 0.0058). In this case, we also observed a significant difference between high-amylose bread and control bread at 45′ (*p* = 0.0371) and 60′ (*p* = 0.0055) (Figure 3). High-amylose bread had a maximum glycaemic peak at 125.4 mg/dL after 45 min from ingestion and the GI was calculated as 90.6 ± 4.19, while common bread showed a maximum peak at 140 mg/dL after 45 min and a GI of 96.82 ± 8.23. These two GIs are significantly different (*p* = 0.0111), in fact the AUC of HA bread was 873.35 ± 60.42 while the AUC of common bread was 931.15 ± 66.03 (*p* = 0.0135).

We also calculated the GL of HA and control bread. In 100 g of HA bread there were 45.8 g of available carbohydrates. Considering a portion of 50 g [37], we found that the GL of HA bread was 20.6 (high glycaemic load), while for the control bread it was calculated as 24.5 (high glycaemic load) because there were 51 g of available carbohydrates in 100 g of product.

### 4.5. Overall Liking and WTP of High-Amylose Biscuits and Taralli

The liking analysis was only performed on biscuits and taralli. The results showed that, although without statistical significance due to the limited sample, the high-amylose biscuits were preferred to the normal biscuits. The average liking score was 6.05 points on 1 to 9 sensory scale, compared to 5.85 points for the control. On the contrary, for the taralli, the average liking score for the control product was higher (6.00 points) than for the high-amylose product (5.20 points).

The WTP values expressed by participants after the blind test did not show any significant differences between the HA and control products. The distribution of the general liking values (still expressed on the 1 to 9 sensory scale) also showed no significant differences.

On the other hand, the values provided by the participants in the second round of WTP, after having received additional information on healthy high-amylose products, showed a significant increase in values. The average bid for high-amylose products was €2.24, statistically different and higher than the average bid for the conventional pack of €1.58. Based on the paired-samples t-test, the two bids were statistically different (*p* < 0.001), and the high-amylose package received an average premium of €0.66. These figures were also confirmed by the answers to the last WTP question, concerning purchase intention for HA products and willingness to recommend the product to others. More than 60% of the observations showed a positive intention to buy HA bakery products, and more than 80% stated that they would recommend HA products to other people.

## 5. Discussion and Conclusions

The potential favourable implication for health given by a higher intake of RS has been extensively discussed in the literature. Indeed, RS in substitution of digestible starch may provide higher satiety, an improvement in gut health, a potential reduction in the risk of colon cancer and diabetes onset, thus sharing positive physiological action with dietary fibre [20]. Normally, the resistant starch content in cereals is negligible (<1%) since starch is rapidly digested in the gastrointestinal tract. On the other hand, high-amylose wheats have the intrinsic properties to provide a higher proportion of resistant starch due to the molecular conformation of amylose and limited granule swelling that make it less susceptible to enzymatic attack [19].

This opens future perspectives to enhance the nutritional properties of cereal-based foods, by providing additional non-digestible carbohydrates in both refined and wholegrain foods. Further, high-amylose products may also have a distinctive tract for consumers, potentially bearing a health claim on resistant starch. We conceived this study with the aim to evaluate the glycaemic response in healthy volunteers after the ingestion of three types of bakery products that can be consumed in each meal to cover the total daily amount of carbohydrates. We used a high-amylose bread wheat flour [32] where a proportion equal to 17% of digestible starch was replaced by resistant starch.

The use of a high-amylose flour to formulate biscuits, taralli and bread determined a substantial increase of resistant starch in the final product compared to controls (<1%). The average amount of total starch replaced by resistant starch was approximately 12%, evidencing a coherence among the three products. Since flour was the only starch-containing ingredient, it is evident by the data expressed on a total starch basis that processing caused an average reduction of 28% in resistant starch in the products. These results can be attributed to the baking phase and starch gelatinization. When starch is heated in a high-moisture environment, macropolymers undergo an irreversible structural change associated with higher viscosity and an open and more flexible structure, developing a higher susceptibility to enzymatic hydrolysis [38].

On the other hand, in control taralli and bread, the higher resistant starch proportion compared to raw material is possibly due to a stronger tendency to retrogradation of amylose chains over cooling [28,39]. According to labelling requirements (Reg. EU 432/2012) [21], none of the products can bear a health claim on resistant starch.

The data related to the glycaemic values in our study population evidenced that HA biscuits and bread resulted in a lower glycaemic index and load than their control counterparts, even if they remained still high and medium, respectively, according to the GI and GL classifications [7,12]. Non-significant differences were observed between HA and control taralli. We also observed a better glycaemic control only after the ingestion of HA bread at 45′ (*p* = 0.0371) and 60′ (*p* = 0.0055) compared to control bread. In the literature, some authors evaluated the effects of HA products both on glycaemic and insulinemic response. Belobrajdic et al. evaluated the effect of four different kind of breads (low- and high-amylose refined bread and low- and high-amylose wholemeal bread) on 20 healthy subjects and found that HA products determined a lower glycaemic and insulinemic response [24]. On the contrary, Hallström et al. in their study on 14 healthy volunteers observed that HA bread provided a lower glycaemic but not insulinemic response after the consumption of high-amylose (38%) bread [30]. HA products provided a slowly digestible starch fraction that determined a reduced absorption of glucose and, consequently, a lower glycaemic response [30].

The study of Corrado et al. (2022) evaluated in vivo glycaemic response and satiety after consumption of a meal consisting of bread rolls formulated with high-amylose flour (SbeIIa mutant, 6.4% resistant starch over total starch). Despite no statistically significant result, due to the limited sample size, they observed a lower glycaemic response, consistent with in vitro observation [40].

Regarding biscuits, the study of Luhovyy et al. evaluated three biscuits with three different concentrations of high-amylose maize flour (1 = control with 100% wheat flour; 2 = low dose with 63% HA flour and 37% maize flour and 3 = high-dose with 33% HA flour and 67% maize flour), but they provided different amounts of available carbohydrates (53.5, 43.5 and 36.3 g, respectively). They observed a 22% lower AUC after the ingestion of the high dose treatment (*p* < 0.05). They concluded that HA maize flour determined a better glycaemic control [41]. On the contrary, in our study both experimental and control products provided 50 g of available carbohydrates.

Our results confirm those of other studies about the better glycaemic control of bread and biscuits [24,42,43]. Moreover, we calculated also the GI that was lower in the two HA products compared to their common counterparts. Currently, in the literature there are no studies on taralli, so our study was the first that evaluated this traditional Italian savoury snack. Nevertheless, HA taralli showed a lower GI than control taralli, but we did not observe significant differences. This may be due to the effects of processing and starch modification, despite resistant starch proportion being comparable with bread and biscuits. In taralli production, after shaping, the dough is boiled in water just before baking. This step is crucial for starch gelatinization resulting in the disruption of starch granules, with increased viscosity and enzymatic accessibility [38], possibly leading to mitigation effects on the ability of amylose in modulating glycaemic response.

Other authors, instead, evaluated the effects of high-amylose flours derived from cereals different from wheat (barley [42,44], rice [45] and corn [46,47],) on glycaemic and insulinemic control. Regarding high-amylose barley, Keogh et al. observed, in 14 women, a significant reduction in the glycaemic and insulinemic curve (22% and 32% respectively with a *p* value of 0.05 in both cases) compared to control products made of HA wheat [42]. Granfeldt et al. in a previous study had already evaluated the effects of barley kernels compared to barley flours on plasma glucose. They observed that intact kernels determined a lower blood glucose rise and that high-amylose barley increased the glycaemia less compared to normal barley and white bread [44]. Goddard et al. conducted a study to test three different kinds of rice containing 0%, from 14 to 17% and from 23 to 25% of amylose to glycaemic and insulinemic response for 180 min. They noticed that after 30 and 60 min both glycaemic and insulinemic responses were lower in the high-amylose kinds of rice compared to 0% amylose rice. High-amylose rice determined also the reduced absorption and digestion of carbohydrates [45]. The insulinemic effect was observed also in a previous study conducted by Behall et al. on 14 subjects affected by hyperinsulinemia. In this case the consumption of high-amylose corn starch normalized the insulinemic response (*p* < 0.002) after 10 weeks, but without significant glycaemic differences compared to the control group (*n* = 10) [46].

Moreover, the consumption of high–amylose seems to be associated to the so–called “second meal effect”. In fact, consuming less digestible carbohydrates may decrease the glycaemic response to a subsequent meal. Ten healthy subjects who consumed high-amylose starch at breakfast showed a lower glycaemic response to a lunch compared to high–amylopectin breakfast [48]. Despite the non-representative sample and the low number of observations, the analysis of sensory liking and WTP showed interesting results consistent with the specific literature [49]. No significant differences were found between HA and control products with regard to liking or the specific sensory attributes used for the panel analysis of bakery products. The WTP results showed a very high premium price, representing approximately 40% of the average value recorded for the ordinary product; this result seems to confirm the sensitivity of consumers to the health aspects associated with snack consumption. In agreement with several scholars [50,51], the liking analysis suggests that HA products may be well accepted by consumers and that a positive willingness to pay can be expected.

To date, managing the glycaemic response after carbohydrate–rich food consumption is one of the major health challenges, especially for subjects that suffer for DMT2 or alteration of glucose metabolism. To the best of our knowledge, this is the first study that evaluated the effect of three high-amylose bakery products on glycaemia. As a strength of this study, we conducted it both on experimental and control food produced with a soft wheat flour chemically similar to the experimental flour, but with a different amount of resistant starch. As a limitation, we did not evaluate the insulinemic response and the satiety after the consumption of these three products. In terms of satiety, some studies in the literature showed that HA wheat products did not affect satiety [24,42]; in fact, in one case, food intake after ingestion was even higher than after the control product [42]. On the contrary, Granfeldt et al. showed that HA barley determined higher satiety compared to white bread and normal barley [44].

As high GI and GL diets have been associated to the onset of T2D, these two parameters may be included in the food labels to make consumers aware of the foods’ characteristics [13]. Indeed, there is actually no consensus among countries in the utilization of GI and GL on food labels [6].

In conclusion, the employment of high-amylose flour contributed to a significant improvement of the nutritional profile of bakery products. Indeed, on average, 12% digestible starch is replaced by resistant starch. Nonetheless, it emerged that 28% of resistant starch lost its indigestibility properties due to processing. High-amylose flour, due to the partial replacement of total starch with resistant starch, determined a lower glycaemic index in both biscuits and bread and better glycaemic control. Further studies are needed to find out the correct formulation or processing management of HA products (especially for taralli) in order to have beneficial effects for subjects affected by diabetes or those at risk.

## Figures and Tables

**Figure 1 foods-12-00319-f001:**
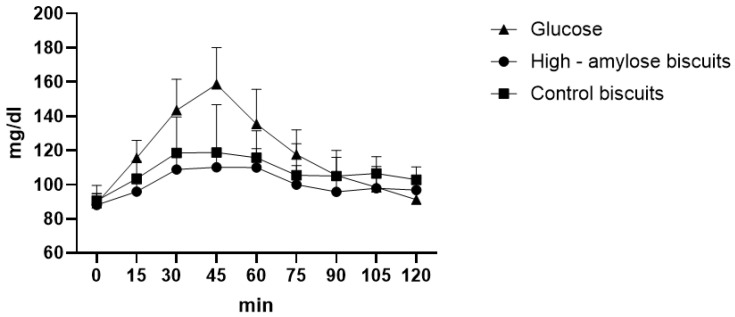
Biscuits glycaemic curve.

**Figure 2 foods-12-00319-f002:**
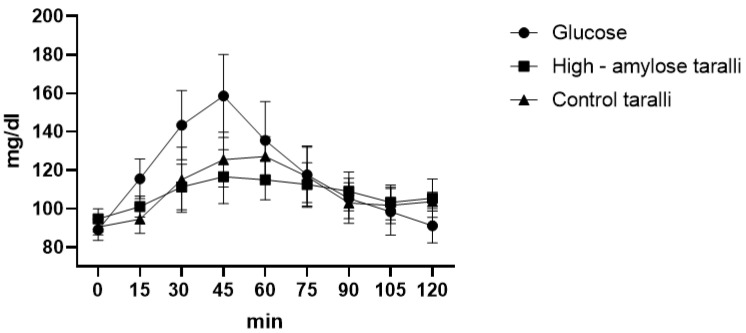
Taralli glycaemic curve.

**Figure 3 foods-12-00319-f003:**
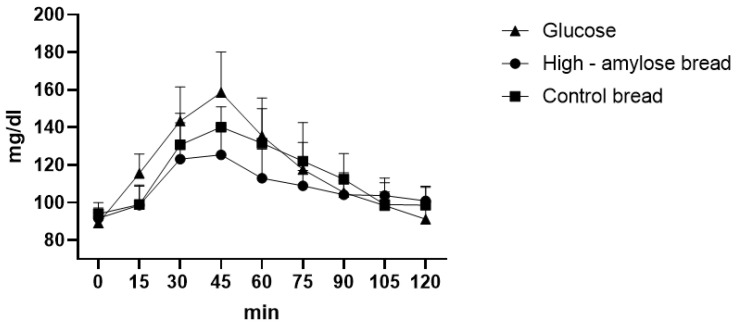
Bread glycaemic curve.

**Table 1 foods-12-00319-t001:** Chemical composition of flours (mean value ± sd).

	Moisture (g/100 g)	Ash (g/100 g d.w.)	TS (g/100 g d.w.)	Fat (g/100 g d.w.)	Protein (g/100 g d.w.) N × 6.25	RS (g/100 g d.w.)	RS (% TS)	Amylose (%)
**HA flour**	14.3 ± 0.1	0.60 ± 0.01	75.2 ± 0.50	2.5 ± 0.18	13.8 ± 0.09	12.7 ± 0.85	16.9	54.4 ± 0.50
**Control flour**	12.8 ± 0.1	0.60 ± 0.04	74.1 ± 1.18	2.4 ± 0.18	11.9 ± 0.02	0.7 ± 0.08	0.9	25.3 ± 0.64

Legend: TS: Total Starch; RS: Resistant Starch.

**Table 2 foods-12-00319-t002:** Mean anthropometrical parameters of participants at baseline.

Parameters	Values
Age (yrs ± SD)	25.2 ± 1.69
Female (%)	50
Height (m ± SD)	1.73 ± 0.098
Weight (kg ± SD)	64.85 ± 11.34
BMI (kg/m^2^ ± SD)	21.45 ± 2.16

**Table 3 foods-12-00319-t003:** Chemical composition (% f.w.) of high-amylose and control bakery products.

	Ash (g/100 g)	Total Starch (g/100 g)	Fat (g/100 g)	Protein (g/100 g) N × 6.25	Available Carbohydrates (g/100 g)	RS (g/100 g)	Sugars * (g/100 g)	RS (% TS)	RS (% Available Carbohydrates)
**High-amylose biscuits**	1.47 ± 0.01 a	39.4 ± 0.91 b	18.4 ± 0.36 a	9.0 ± 0.01 a	58.3	4.6 ± 0.40 a	18.9	11.7	7.9
**Control biscuits**	1.44 ± 0.04 a	43.2 ± 0.65 a	17.0 ± 0.37 b	8.9 ± 0.01 b	61.9	0.2 ± 0.05 b	18.7	0.4	0.6
**High-amylose taralli**	1.45 ± 0.04 A	55.0 ± 1.57 A	16.3 ± 0.27 A	10.5 ± 0.00 A	55.0	6.9 ± 0.42 A	-	12.5	12.5
**Control taralli**	1.42 ± 0.02 A	55.8 ± 1.20 A	17.5 ± 0.45 A	9.0 ± 0.09 B	55.8	0.7 ± 0.06 B	-	1.3	1.3
**High-amylose bread**	1.17 ± 0.04 β	45.9 ± 0.01 β	1.3 ± 0.20 α	7.8 ± 0.01 β	45.8	5.7 ± 0.04 α	-	12.5	12.5
**Control bread**	1.34 ± 0.02 α	50.7 ± 0.55 α	1.5 ± 0.13 α	8.7 ± 0.02 α	51.0	1.4 ± 0.54 β	-	2.8	2.8

Legend: * calculated as sugar added in the formulation; RS: resistant starch; TS: total starch; -: not available. Different lower-case, upper-case and Greek letters, within each column, indicate statistically significant differences (*p* < 0.05) between high-amylose and control biscuits, taralli and bread, respectively.

## Data Availability

The data are available from the corresponding author.

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
