# Peer review of "Effect of Three Bakery Products Formulated with High-Amylose Wheat Flour on Post-Prandial Glycaemia in Healthy Volunteers"

_foods, 2023, doi:10.3390/foods12020319_

Round 1
Reviewer 1 Report
This is a very good research. Comments that may improve are within the manuscript.

Author Response
Dear Reviewer 1,
thank you for your comments. Attached you can find the answers to your suggestions.

Reviewer 2 Report
Please see in the attach.

Author Response
Dear Reviewer 2,
thank you for your comments. Attached you can find the answers to your suggestions.

Reviewer 3 Report
This is an interesting manuscript dealing with the effect of three bakery products formulated with high-amylose bread wheat flour on post-prandial glycemia in healthy volunteers.
The abstract should be more result-oriented instead of being method-oriented.
The introduction and method parts are written very well.
In table 3, I think the author should show the significant differences between the samples regarding the chemical composition (with one – way ANOVA and Tukey’s post hoc test as you mentioned in the statistical analysis part).
The discussion part should be improved by comparing of the results with more studies in the field.
For my information, is the production of these products financially and technologically affordable? What are the challenges?
I think that it would have been better if tests such as the textural properties analysis were also performed in addition to the tests done to pay attention to the technological aspect of these healthy products.
In the reference list, some journal titles are mentioned in full form whereas some are in the abbreviated form; please correct them according to the journal format.
Author Response
Dear Reviewer 3,
thank you very much for your comments. Attached you can find the answers to your suggestions.
